# Examining the Effects of Altitude on Workload Demands in Professional Basketball Players during the Preseason Phase

**DOI:** 10.3390/s24103245

**Published:** 2024-05-20

**Authors:** Sergio J. Ibáñez, Carlos D. Gómez-Carmona, Sergio González-Espinosa, David Mancha-Triguero

**Affiliations:** 1Research Group in Optimization of Training and Sports Performance (GOERD), Department of Didactics of Music Plastic and Body Expression, Faculty of Sport Science, University of Extremadura, 10003 Caceres, Extremadura, Spain; sibanez@unex.es (S.J.I.); cdgomezcarmona@unex.es (C.D.G.-C.); sergio.gonzalezespinosa@unir.net (S.G.-E.); 2BioVetMed & SportSci Research Group, International Excellence Campus “Mare Nostrum”, Department of Physical Activity and Sport, Faculty of Sport Sciences, University of Murcia, 30720 San Javier, Murcia, Spain; 3NÌKE: Research Group in Education, Health and Sports Performance, Didactics of Physical Education and Health, International University of La Rioja, 26006 Logroño, La Rioja, Spain; 4Physical Education and Sports Department, Cardenal Spínola CEU, Andalucía University, 41930 Bormujos, Sevilla, Spain; 5Physical Education and Sports Department, Fundación San Pablo CEU, Andalucía University, 41930 Bormujos, Sevilla, Spain

**Keywords:** training programs, hypoxia, heart rate, locomotion, microtechnology, sports

## Abstract

Basketball involves frequent high-intensity movements requiring optimal aerobic power. Altitude training can enhance physiological adaptations, but research examining its effects in basketball is limited. This study aimed to characterize the internal/external workload of professional basketball players during preseason and evaluate the effects of altitude and playing position. Twelve top-tier professional male basketball players (Liga Endesa, ACB; guards: *n* = 3, forwards: *n* = 5, and centers: *n* = 4) participated in a crossover study design composed of two training camps with nine sessions over 6 days under two different conditions: high altitude (2320 m) and sea level (10 m). Internal loads (heart rate, %HR_MAX_) and external loads (total distances covered across speed thresholds, accelerations/decelerations, impacts, and jumps) were quantified via wearable tracking and heart rate telemetry. Repeated-measures MANOVA tested the altitude x playing position effects. Altitude increased the total distance (+10%), lower-speed running distances (+10–39%), accelerations/decelerations (+25–30%), average heart rate (+6%), time in higher-intensity HR zones (+23–63%), and jumps (+13%) across all positions (*p* < 0.05). Positional differences existed, with guards accruing more high-speed running and centers exhibiting greater cardiovascular demands (*p* < 0.05). In conclusion, a 6-day altitude block effectively overloads training, providing a stimulus to enhance fitness capacities when structured appropriately. Monitoring workloads and individualizing training by playing position are important when implementing altitude training, given the varied responses.

## 1. Introduction

Basketball is an intense intermittent team sport characterized by frequent high-intensity movements, including sprints, jumps, shuffles, and rapid changes in direction [1]. These external workload demands impose considerable physiological strain, demonstrated by average heart rate (HR) over 80% of maximum HR and blood lactate levels exceeding 8 mmol/L during competition [2]. A key determinant of a basketball player’s capacity to meet these internal [3] (effect that a certain effort causes in the body based on the task that has been assigned to the athlete) and external [4] (indicator data of the work performed, quantified through parameters such as duration, volume, intensity, etc.) workload requirements is maximal oxygen uptake (VO2_MAX_) [5]. The average VO2_MAX_ values of elite basketball players range from 50 to 60 mL/kg/min, with guards exhibiting slightly higher aerobic capacities than forwards and centers [2]. Greater VO2_MAX_ levels enable faster recovery following repeated high-intensity bursts by enhancing oxygen delivery and buffering accumulating metabolites [6].

Developing VO2_MAX_ in basketball players traditionally utilizes high-intensity interval training under normoxic conditions [7]. However, exposure to hypoxic conditions in simulated or real environments 2000–3000 m above sea level (m.a.s.l.) stimulates physiological adaptations such as increased hemoglobin mass, blood buffering capacity, and oxygen extraction, which enhance oxygen delivery and utilization [8]. In recent years, there has been a shift in altitude training strategy from “living high—training low” to “living low—training high” due to evidence showing better benefits of training high compared to living high for sea-level performance [9,10]. Most altitude research has focused on endurance sports, but implementing altitude training camps during the preseason phase may provide basketball teams with competitive advantages [11]. A few studies in basketball have shown the improvement of aerobic capacities [12], high-intensity running performance [13], and on-court high-intensity actions (sprints, jumps, change in direction, etc.) following hypoxic training [14,15].

Although hypoxic training enhances physical and physiological performance in basketball players, an increase in fatigue and stress values [14], along with reductions in lower limb balance [16] and sleep quality [17], has been reported. Monitoring workloads at altitude is critical to ensure optimal adaptations. External workloads can be quantified using accelerometers, local positioning systems, and time–motion analyses [18]. Internal workload measures like heart rate variability and perceived exertion provide insight into the physiological strain incurred [2]. While previous basketball research evaluated physical fitness outcomes by altitude training, analyses of workload differences between high-altitude and sea-level training environments are lacking. 

In addition, playing positions present different physical demands [19,20] and technical–tactical actions [21] due to their specific roles in the game. Different adaptations to altitude have been identified across playing positions. Concerning internal responses, guards and forwards report higher in-game fatigue and perceive greater exertion, while centers experience more muscular pain and longer recovery times [22,23,24]. In terms of external workload demands, guards cover less distance but with greater intensity, forwards cover moderate total distances with mixed intensities, and centers have lower court time and distance but perform more static actions with contact on the paint [2,19,25].

Therefore, this study aimed to characterize the external and internal workload demands of basketball players during the preseason and evaluate the effects of altitude (training high—living high) in comparison to sea level (training low—living low) on these demands by playing position. It was hypothesized that altitude exposure would produce an effect on physical and physiological demands across playing positions. Investigating the influence of altitude training can help to determine the effectiveness of this protocol for optimizing workload capabilities in preparation for basketball competitions.

## 2. Materials and Methods

### 2.1. Design

A within-subjects, quasi-experimental study following a quantitative and descriptive design was conducted to evaluate the effect of altitude on the training location (sea level vs. high altitude) in a Spanish basketball professional team (Liga Endesa, ACB) during the preseason phase [26]. The data collection of internal and external workload demands was performed by electronic performance tracking systems and heart rate telemetry from 18 training sessions in 12 days (9 sessions at high altitude vs. 9 sessions at sea level). The session structure, training objectives (strength and conditioning, technical and tactical components), and tasks between the two conditions were equal. A 48 h washout period between the high-altitude and sea-level training interventions was implemented to eliminate fatigue and achieve an optimal post-exercise recovery [11]. In addition, the group design was equivalent through the study of the training load [26].

### 2.2. Participants

Fifteen professional basketball players (age: 24.6 ± 6.2 years; height: 196.8 ± 11.1 cm; weight: 87.3 ± 10.2 kg) belonging to a top-tier Spanish league team (Liga Endesa, ACB) participated in the present study. All of the players were active professionals contracted with the same team during the competitive season when the study was conducted. The inclusion criteria were as follows: (a) Belonging to the official squad of the ACB League first team. (b) Participating in all training sessions. (c) Not having suffered a musculoskeletal injury in the 15 days prior to the start of the preseason, which would limit their maximum performance. (d) Having had an adaptation period of 2 days before the technological equipment was equipped, and agreeing to be equipped during training. (e) Participating voluntarily and signing an informed consent form. For these reasons, only 12 players that met the inclusion criteria were included in the final analysis (guards, *n* = 3; forwards, *n*= 5; centers, *n* = 4).

Before the assessments, the basketball team allowed the research team to access the players’ data, and all players provided written informed consent. The study was conducted in accordance with the ethics code of the World Medical Association and the 7th edition of the Declaration of Helsinki [27], and it was approved by the Ethics Committee of the University of Extremadura (protocol code: 233/2019, date of approval: 1 November 2019).

### 2.3. Variables

In this study, training altitude and playing position were considered as independent variables. To assess the impact of altitude, two experimental conditions were established: (a) sea-level training (10 m.a.s.l.), and (b) high-altitude training (2320 m.a.s.l.). The sea-level sessions were completed in the sports center where the professional team usually plays its games during the season. The high-altitude sessions took place at the High-Performance Center in Sierra Nevada, Granada, Spain. Furthermore, in order to individualize the effects of the training process and environmental conditions concerning the technical–tactical characteristics of athletes, playing position was included as an independent variable and was categorized into three groups: (a) guards, (b) forwards, and (c) centers.

On the other hand, internal and external workload demands were registered as dependent variables to analyze the effects of altitude. Variables were divided into four categories: *Locomotion*: This represents the displacements realized by the players, and it was measured in meters/minute—relative distance (RD)—and at different intensities (walking, 0–6 km/h; jogging, 6–12 km/h; running, 12–18 km/h; high-intensity running (HIR), 18–21 km/h; sprinting, 21–24 km/h; and maximum sprinting, >24 km/h).*Speed and speed changes*: This represents the velocity of displacements and the positive and negative changes on them. The velocity of displacements was measured in km/h with two variables: average speed (Speed_AVG_) and maximum speed (Speed_MAX_). On the other hand, speed changes were measured by total accelerations (TAcc) and decelerations (TDec), in counts/min; maximum acceleration (Acc_MAX_) and deceleration (Dec_MAX_), in m/s^2^; and relative distance covered in acceleration (RD_Acc_) and deceleration (RD_Dec_), in meters/minute. The threshold to detect accelerations and decelerations was positive or negative changes in speed of 0.1 m/s^2^, respectively.*Neuromuscular load*: This represents the impact of displacements on the muscle body concerning the force of gravity. The measured variables were as follows: Player load, measured by RealTrack Systems (PL_RT_) in a.u./min; total impacts and at different intensities (low, 0–3 g; moderate, 3–5 g; high, 5–8 g; very high, >8 g); and steps and jumps, in counts/min. PL_RT_ was represented in arbitrary units (a.u.) and was calculated directly by the manufacturer’s software (SPRO^TM^, version 989, RealTrack Systems, Almeria, Spain) using the following equation, at a sampling frequency of 100 Hz, where PLn is the PL calculated at the current instant in time; n is the current instant in time; *n* − 1 is the previous instant in time; *X_n_*, *Y_n_*, and *Z_n_* are the values of body load in each axis of movement at the current time; and *X_n_*_−1_, *Y_n_*_−1_, and *Z_n_*_−1_ are the values of body load in each axis of movement at the previous instant in time [27]:
$$PL_{n}=\sqrt{\frac{\;{\left(X_{n}-X_{n-1}\right)}^{2}+\;{\left(Y_{n}-Y_{n-1}\right)}^{2}+\;{\left(Z_{n}-Z_{n-1}\right)}^{2}}{100}\;}$$
$$PL_{RT}=\overset{m}{\underset{n=0}{\sum }}PL_{n}\times 0.01\;$$*Heart rate telemetry*: This represents the physiological effect of external load. It was measured by maximum heart rate (HR_MAX_) and mean heart rate (HR_AVG_) in beats per minute (bpm), as well as in the percentage of time spent at different intensities (very low, 50–60% HR_MAX_; low, 60–70% HR_MAX_; moderate, 70–80% HR_MAX_; high, 80–90% HR_MAX_; very high, 90–95% HR_MAX_; and maximum, 95–100% HR_MAX_). Data were extracted at the end of each session. To determine individual HR_MAX_ percentages, each player’s HR_MAX_ was established using previously gathered data from laboratory evaluations conducted under the supervision of the medical team staff before the start of the assessment.

### 2.4. Equipment

WIMU PRO inertial devices (RealTrack Systems, Almería, Spain) were used for the data collection. The devices were attached to the players with an anatomical vest to ensure no movement during registers, and at the inter-scapulae level due to this being the best place for tracking detection [28]. The participants’ locomotion, speed, and speed changes were monitored using ultra-wideband (UWB) technology to enable device-based tracking of basketball players indoors. UWB referencing of devices on the court was accomplished utilizing 500 MHz radiofrequency technology and 33 Hz data collection through eight anchors pre-positioned around the perimeter. The installation protocol and the precision of this UWB tracking system have been validated in prior studies [29]. 

Neuromuscular load variables were registered through the inertial sensors that composed the WIMU PRO devices (4 accelerometers ±16, ±16, ±32, and ±400 g; 3 3D gyroscopes with a range output of ±2000 °/s, ±2000 °/s, and ±4000 °/s; a magnetometer), at a sampling frequency of 1000 Hz. The validity and reliability of these sensors have been evaluated previously, with satisfactory results [28]. Finally, heart rate telemetry was registered with a GARMIN band linked with the WIMU PRO inertial device. 

### 2.5. Procedures

First, a request was made to the professional basketball team to avoid data collection. Once the proposal had been accepted, informed consent was signed, and a familiarization session with high monitoring was conducted to familiarize the players with the equipment used in the study. During the familiarization session, anthropometric data were obtained to characterize the sample through a rod stadiometer (SECA, Hamburg, Germany) and a body composition monitor (Model BC-601, TANITA, Tokyo, Japan), and data on birth date and playing position were obtained with a survey completed by the team staff.

Data collection for this study occurred in two phases over a total of 12 days: (a) training high—living high and (b) training low—living low. First, the participants completed nine training sessions over six days at the High-Performance Center in Sierra Nevada, Granada, Spain (2320 m.a.s.l.). This high-altitude camp was followed by a second phase where players completed nine training sessions over six days in their home sports center (10 m.a.s.l.). The participants resided and trained at the assigned altitude for the two weeks of the training intervention. Between the two interventions, a 48 h washout period was performed for post-exercise recovery [11].

Each training session had a total duration of 75 to 90 min. The 20 min standardized warm-up was composed as follows: (1) 10 min of mobility, balance, and dynamic stretching; (2) 5 min of jumping/plyometrics, and changes in speed and direction on the court; and (3) 5 min of technical individual drills (bounds, throws, etc.). After the warm-up, 45 to 60 min of main training was conducted following the session objectives. The description of the training drills was realized following the nomenclature provided by [30] (Figure 1). For the last 10 min of each session, players performed a cool-down period based on free, 2-point, and 3-point throws and individual recovery protocols (e.g., foam roller, static stretching). Additionally, 30 min before the start, players were instructed on the court to place the WIMU PRO inertial devices on the anatomical vests, as well as to carry out individual exercises with the foam roller and with the ball to prepare the neuromuscular system to start the session. The usual verbal encouragement from the head coach was allowed during sessions. In addition, players had ad libitum access to water and energetic supplementation during recovery periods.

After each data collection, the sensor data files generated by the WIMU PRO units were extracted on a laptop with the Windows operating system and analyzed using the manufacturer’s software. SPRO software (version 990, RealTrack Systems, Almeria, Spain) processes the raw data of the inertial device sensors into quantitative biomechanical metrics, including both native sensor outputs and derived parameters calculated through sensor fusion algorithms. The quantified metrics that provide objective measures of the participants’ physical activities and movements were extracted to create the database on Excel software (Version 2205, Microsoft, Redmond, WA, USA) for further analyses.

### 2.6. Data Analysis

Firstly, a descriptive analysis of the external and internal workload variables in each condition (high altitude vs. sea level) was performed to obtain information as the mean ± standard deviation. Subsequently, an exploratory analysis was performed using the criteria assumption tests for normality with the Kolmogorov–Smirnov test and homoscedasticity with Levene’s test, so parametric tests were performed. A repeated-measures MANOVA test was used to analyze the effects of altitude (high altitude vs. sea level) × playing position (guard, forward, or center) on the external and internal workload variables of basketball players, using the Bonferroni post hoc test for pairwise comparisons. To analyze the magnitude of differences, the partial omega squared (*ω_p_*^2^) was used for MANOVA effects and interpreted as follows [31]: *ω_p_*^2^
*>* 0.01 small, *ω_p_*^2^
*>* 0.06 moderate, *ω_p_*^2^
*>* 0.14 large; while Cohen’s d was used for post hoc effects and interpreted as follows [31]: *d* > 0.20 small, *d* > 0.50 moderate, *d* > 0.80 large. The software used for the statistical analysis was JAMOVI (Version 2.4.1, The Jamovi Project, Sydney, Australia), while graphs were designed by GraphPad Prism (release 8, GraphPad Software, La Jolla, CA, USA). The significance value was established at *p* < 0.05.

## 3. Results

### 3.1. Effects of Altitude on Internal and External Workload Demands

Figure 2 shows the results of the effects of altitude on internal and external workload demands in professional basketball players during the preseason phase. Firstly, no differences were found in total training time between the two conditions (high altitude = 87.2 ± 15.3 min; sea level = 86.6 ± 12.8 min; *p* = 0.87, *d* = 0.02). Regarding the comparison of internal and external workload demands, higher demands were found at high altitude, with high effect size in the relative distance 0–6 km/h, total accelerations/decelerations, Acc_MAX_, Dec_MAX_, and relative distance in acceleration and deceleration; with moderate effect size in total distance, HR_AVG_, and 80–90% HR_MAX_; and with low effect size in the relative distance 6–12 km/h, 70–80% HR_MAX_, 90–95% HR_MAX_, 95–100% HR_MAX_, and total jumps. Higher demands at sea level were only found in 50–60% HR_MAX_, with a low effect size.

### 3.2. Combined Effect of Altitude and Playing Position

The descriptive analysis and repeated-measures MANOVA statistics to analyze the effect of altitude x playing position are shown in Table 1. No interactions were found between the two variables. Concerning playing position, differences were found, with moderate effect size in relative distance 12–18 km/h (guards > centers; forwards > centers), impacts 3–5 g (guards > forwards/centers), and impacts 5–8 g (guards > forwards/centers); and with low effect size in relative distance 0–6 km/h (centers > guards/forwards), HR_AVG_ (centers > forwards), 50–60% HR_MAX_ (forwards > guard/-centers), 80–90% HR_MAX_ (guards/centers > forwards), 90–95% HR_MAX_ (centers > guards/forwards), Speed_AVG_ (guards/forwards > centers), impacts >8 g (guards > forwards/centers), total steps (guards > forwards/centers), PL (guards > forwards/centers), and relative distance in acceleration and deceleration (guards > forwards/centers).

### 3.3. Specific Effects of Altitude by Playing Position

Figure 3 shows the independent effects of altitude in each playing position. Concerning the total distance covered at different speeds, all differences were found to have higher values for the high-altitude condition in RD and RD_0–6 km/h_ (high effect: guards; moderate effect: forwards and centers), RD_12–18 km/h_ (moderate effect: guards), RD_18–24 km/h_ (high effect size: all positions), and RD_>24 km/h_ (moderate effect: centers; low effect: forwards).

Regarding the internal demands according to heart rate telemetry, higher demands were found at sea level in 50–60%HR_MAX_ (high effect: guards; moderate effect: centers; low effect: forwards) and 60–70%HR_MAX_ (moderate effect: centers). On the other hand, higher demands at high altitude were obtained in HR_AVG_ (moderate effect: guards and centers; low effect: forwards), 70–80%HR_MAX_ (moderate effect: guards and centers; low effect forwards) 80–90%HR_MAX_ (high effect: guards; moderate effect: centers; low effect: forwards), and 90–95%HR_MAX_ and 95–100%HR_MAX_ (high effect: centers).

In terms of speed and speed changes, a high effect size was found for the high-altitude condition in all playing positions in TAcc, TDec, Acc_MAX_, Dec_MAX_, RD_Acc_, and RD_Dec_. Finally, in terms of neuromuscular load, only centers presented moderate effect on jumps.

## 4. Discussion

Aerobic capacity is a determinant of the physical and physiological performance of basketball players [5]. For its development, high-intensity interval training has been used under normoxic conditions [7], but implementing training camps in hypoxic conditions (training high—living high) could provide higher performance enhancements [8]. Due to basketball research not evaluating the direct effects of altitude on workload demands during a training camp, this study aimed to characterize external and internal load demands during preseason and examine differences based on the training environment (sea level, 10 m.a.s.l. versus high altitude, 2320 m.a.s.l.) by playing position. Overall, hypoxic exposure increased physical and physiological demands, aligning with this study’s hypothesis.

### 4.1. Effects of Altitude on Internal and External Workload

The results indicated that internal and external workload demands were effectively overloaded in the 6-day altitude camp compared to the sea-level camp across all positions, as evidenced by heightened locomotor activities and cardiovascular strain. Specifically, total distance and at low intensity (<12 km/h), accelerations/decelerations (total, maximum efforts, and distance covered), and jump frequency were 10–39% higher at altitude. Concerning the effect of total distance and at lower intensities (TD_0–6 km/h_ and TD_6–12 km/h_), previous research in soccer has shown increased low-speed running but reductions in high-speed running early at altitude, as players work harder to maintain their movement velocities due to the decreased partial pressure of inspired oxygen [32]. In this sense, [33] found that an altitude > 1500 m improved the best marks of sprinting (<400 m), triple and long jump, and hammer throw, due to the reduction in air density and aerodynamic drag, meaning less resistance for fast movements. Therefore, the altitude environment directly influences the external workload of athletes, with higher performance in speed changes and jumps but lower performance in total distance covered at high intensities, which depends on the aerobic metabolic energy system.

Greater physiological demands were illustrated by higher average heart rates and more time spent at moderate–very-high-intensity zones during altitude sessions. HR data may be impacted by the geographical location, specifically at higher altitudes [33] and in simulated conditions [16]. HR remains elevated after 3–5 days after initial exposure to altitude, compensating for a decrease in stroke volume [34]. Also, training high and living high include the potential for high-altitude mountain sickness (AMS) caused by reduced air pressure and lower oxygen levels. This is characterized by headaches, nausea, breathlessness, vomiting, and dizziness [35], which affect fatigue and stress values [33], lower limb balance [16], and sleep quality [17].

When the athletes returned to sea level, an increased efficiency of displacements was found, with a reduction in total distance and at low intensity, speed changes, and jumps, as well as maintaining or increasing the high-intensity distance. This efficiency and higher performance are in agreement with other basketball research reporting improved high-intensity running capacity [13] and on-court agility performance [12,14] following hypoxic training. In addition, the increased aerobic demands under hypoxia concur with past studies documenting superior VO2_MAX_ development [11] and faster heart rate recovery [34] in basketball players after altitude training. The authors of [15] found that HIIT in hypoxia improved maximum aerobic capacity more than HIIT in normoxia.

Other methods that could be investigated that increase the physical and physiological performance at sea level include living high—training low, training high—living low, and training low with a reduction in breathing rate or air volume. Firstly, not only did training at real or simulated altitudes improve performance, living high—training low also produced improvements in repeated sprint ability and aerobic capacity [36]. In a comparison between the benefits of living high and training high, training high presented greater benefits specifically in normobaric hypoxia, due to maintaining the physiological benefits of oxygen reduction and not altering the neuromuscular system [9]. Hypobaric hypoxia (real altitude) produces higher heart rate, decreased minute ventilation and alveolar ventilation, higher Lake Louise AMS (headache, nausea, fatigue, dizziness), and worse balance compared to normobaric hypoxia [37]. Finally, new methods with lower costs than living high or training high include hypoventilation training and reduction in air volume, which have shown an increase in VO2_MAX_, being adaptable to all sports and all conditions [14].

Therefore, the use of short 5–7-day altitude blocks during preseason can provide an overload stimulus to develop physical and physiological fitness across all positions, so long as workloads are monitored to manage the fatigue and stress produced by altitude exposure. Upon returning to sea level, players displayed improved efficiency and performance, suggesting the utility of hypoxic exposure for enhancing high-intensity running capacity and on-court agility. While living high—training high is an effective method, alternatives like simulated hypoxia, reduced air volume (masks), and hypoventilation could be accessible methods for varied sports and conditions, with a reduced cost. Finally, the inclusion of more altitude training periods within the annual plan can optimize performance gains after tapering at sea level.

### 4.2. Specific Demands between Playing Positions

While an altitude effect occurred across all playing positions, the workload magnitudes differed between guards, forwards, and centers. Playing positions present different roles during the game that affect physical [19,20] and technical–tactical actions [21]. Guards covered more distance in high-speed running, change in direction, and acceleration/deceleration versus forwards and centers. These results align with time–motion analyses demonstrating that guards complete more high-intensity activities during games [20,25]. Centers accrued less distance in moderate/high-speed locomotion but exhibited greater cardiovascular demands [24]. The positional differences under hypoxia concurred with past research showing varied physical and physiological adaptations to altitude between players [13]. 

Examining workload responses separately by position revealed that altitude universally increased demands, although the magnitudes differed. All positions showed large improvements in high-speed running, total number of accelerations/decelerations, and distance covered. Guards demonstrated the greatest altitude gains in high-intensity cardiovascular strain, highlighted by very high (80–100% max HR) zones. These results align with the findings of [13], who reported enhanced aerobic fitness in guards following altitude training. Centers experienced high heart rate elevations at lower intensities (50–70% max HR) during hypoxic sessions, concurring with [23], who found that centers reported greater muscular pain and longer recovery times. Positional data provide novel evidence that hypoxic exposure stimulates greater speed/agility volumes and cardiovascular overload across playing positions.

Therefore, basketball players in different positions respond uniquely to altitude training, although all experience overload. Guards may focus on developing speed and agility, while big men emphasize lower-speed endurance. Coaches should optimize training by considering each position’s strengths and weaknesses and tailoring the altitude stimulus accordingly. For example, increased high-intensity running for guards provides a platform to improve their speed endurance. Monitoring workload helps balance the desired adaptations with fatigue. Tapering after altitude should align with positional requirements as players transition back to sea level. Field testing after altitude helps assess individual progress to inform further training needs at sea level.

### 4.3. Limitations

Although the present study provides the first insight into the effects of training environment (high altitude vs. sea level) on internal and external workload demands by playing position in a professional basketball team, several limitations should be mentioned. The results may not be broadly generalizable beyond the single elite team examined. Implementing longer altitude exposures across a larger, more diverse sample could improve the generalizability and provide additional workload insights over an extended timeframe. The logistical complexities and costs of traveling to appropriate altitude facilities may limit accessibility for many teams. Evaluating alternative hypoxic training methods could offer more practical solutions. This study was confined to the preseason period; investigating altitude applications during other phases may reveal new timing effects. Finally, a longitudinal approach tracking players across successive altitude camps could better assess individual adaptations and performance improvements over time. Future lines of research could also analyze different training periods (short periods of training at altitude and high periods of training at altitude). In addition, it could also be interesting analyze aerobic and training resistance together with strength performance.

## 5. Conclusions and Practical Applications

This study provides novel insights into the influence of altitude training on basketball workload demands during the preseason. The results showed that exposure to 2320 m meaningfully increased external loads, including total distance, accelerations/decelerations, and high-intensity running. Internal loads were also greater at high altitude, evidenced by heightened cardiovascular strain. The magnitude of the difference between high altitude and sea level was similar for guards, forwards, and centers. A 6-day altitude camp effectively overloads basketball training, which could potentiate fitness adaptations when structured appropriately within the annual plan.

Therefore, due to data indicating that integrating small altitude camps into preseason enhances workload capacities, different practical applications could be given for basketball team staff: (a) using short altitude blocks (5–7 days) during early preseason to provide an intensive overload stimulus, drive training adaptations, and improve the group cohesion; (b) workload monitoring is key to balancing the desired physical fitness adaptations during altitude training, controlling stress and fatigue; (c) coaches should individualize training at altitude based on position-specific demands; (d) blending technical/tactical activities with fitness loads helps develop skills under hypoxic conditions; and (e) a tapering period at sea level after altitude training may optimize performance gains.

## Figures and Tables

**Figure 1 sensors-24-03245-f001:**
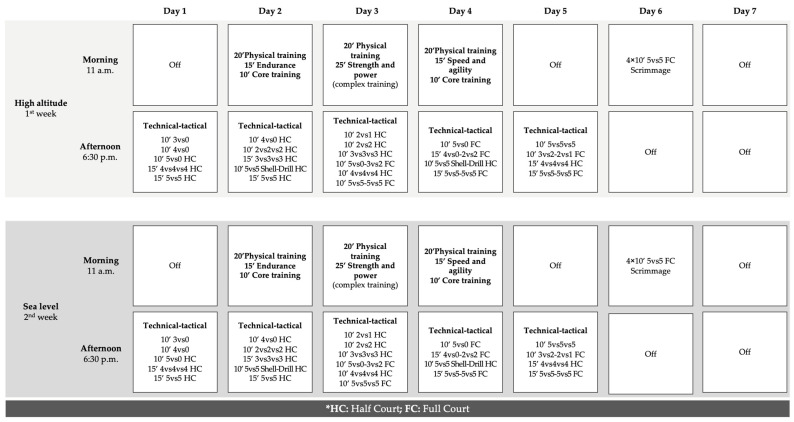
Study design, weekly training structure, and contents for both conditions.

**Figure 2 sensors-24-03245-f002:**
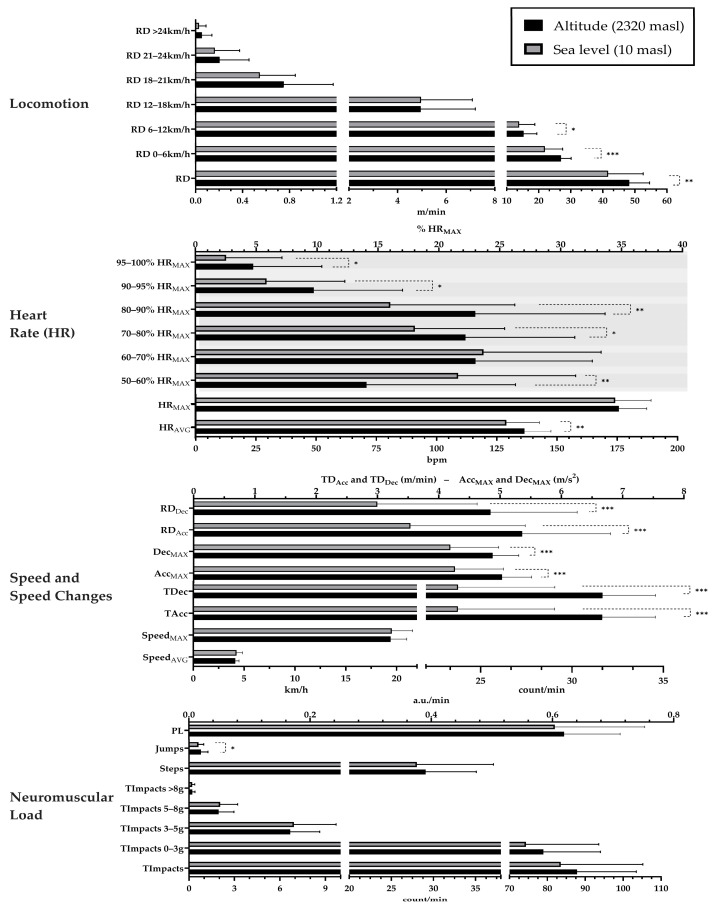
Bar plot that represents the effects of altitude on internal and external workload demands in professional basketball players. **Note:** * low effect size (*d =* 0.20–0.50); ** moderate effect size (*d =* 0.50–0.80); *** high effect size (*d >* 0.80).

**Figure 3 sensors-24-03245-f003:**
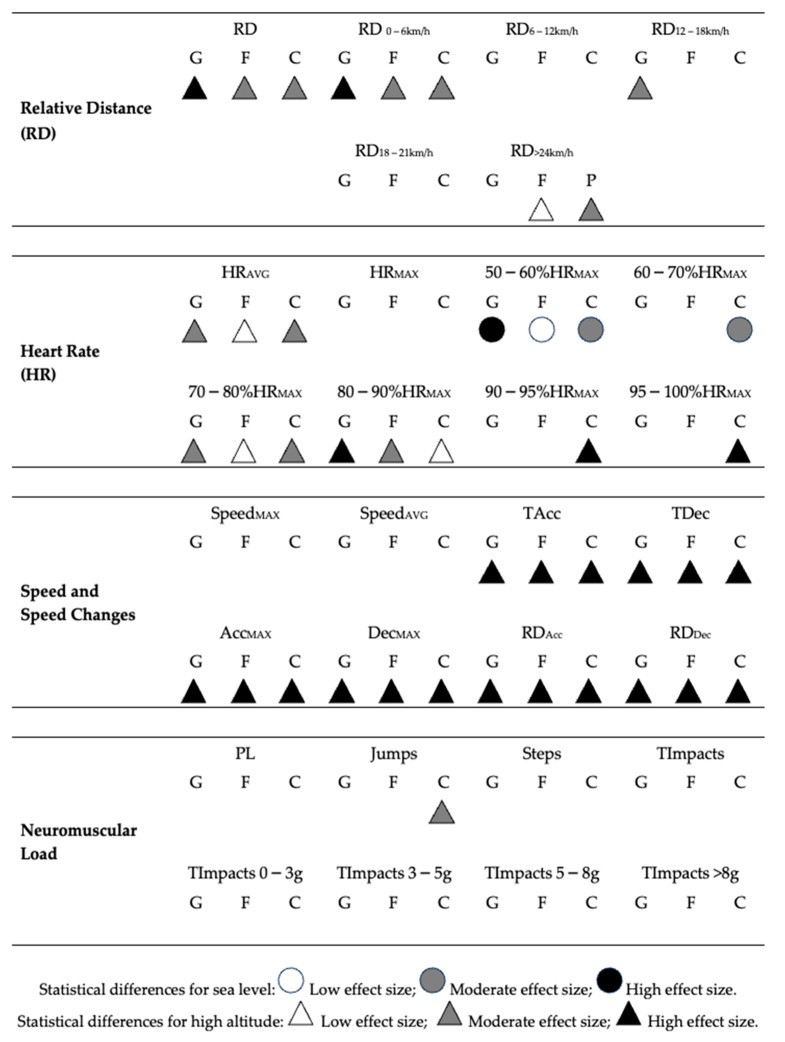
The specific effects of altitude on internal and external workload demands, by playing position. **Note**: G: guards; F: forwards; C: centers; HR: heart rate; PL: player load; RD: relative distance.

**Table 1 sensors-24-03245-t001:** MANOVA playing position vs. altitude in basketball players during the preseason phase.

Variables	Altitude	Playing Position			*F* (*p-Value*)	*ω_p_*^2^** (*Rating*)	*Post Hoc*
		Guard	Forward	Centers			
**RD (m/min)**	2320 masl	50.84 ± 8.30	48.03 ± 6.23	47.36 ± 5.28	0.70 (0.49)	0	
	10 masl	40.16 ± 12.84	42.35 ± 9.52	41.74 ± 12.01			
	*F (p)*	**23.35 (<0.01)**			Interaction *F (p value); ω_p_*^2^* (rating)*		
	*ω_p_*^2^* (rating)*	**0.11 (moderate)**			0.50 (0.61); 0		
**RD_Walking_** **(0–6 km/h)** **(m/min)**	2320 masl	26.68 ± 3.24	26.40 ± 3.28	27.91 ± 2.83	**2.97 (0.05)**	**0.01 (trivial)**	**e f**
	10 masl	20.46 ± 6.25	21.93 ± 4.74	23.77 ± 6.04			
	*F (p)*	**41.98 (<0.01)**			Interaction *F (p value); ω_p_*^2^* (rating)*		
	*ω_p_*^2^* (rating)*	**0.18 (high)**			0.27 (0.77); 0		
**RD_Jogging_** **(6–12 km/h)** **(m/min)**	2320 masl	17.05 ± 4.70	15.44 ± 3.72	14.33 ± 4.20	1.99 (0.14)	0	
	10 masl	13.99 ± 5.43	14.14 ± 4.58	13.21 ± 5.14			
	*F (p)*	**5.50 (0.02)**			Interaction *F (p-value); ω_p_*^2^* (rating)*		
	*ω_p_*^2^* (rating)*	**0.02 (low)**			0.26 (0.77); 0		
**RD_Running_** **(12–18 km/h)** **(m/min)**	2320 masl	6.33 ± 2.23	5.17 ± 2.06	4.01 ± 2.17	**7.53 (<0.01)**	**0.06 (moderate)**	**b d**
	10 masl	4.82 ± 2.13	5.39 ± 2.03	4.09 ± 2.08			
	*F (p)*	1.23 (0.27)			Interaction *F (p-value); ω_p_*^2^* (rating)*		
	*ω_p_*^2^* (rating)*	0			1.70 (0.19); 0		
RD_HIR_ (18–21 km/h) (m/min)	2320 masl	0.62 ± 0.35	0.70 ± 0.71	0.88 ± 0.86	0.08 (0.93)	0	
	10 masl	0.67 ± 0.56	0.68 ± 0.48	0.54 ± 0.47			
	*F (p)*	1.01 (0.32)			Interaction *F (p-value); ω_p_*^2^* (rating)*		
	*ω_p_*^2^* (rating)*	0			1.51 (0.22); 0		
RD_Sprinting_ (21–24 km/h) (m/min)	2320 masl	0.14 ± 0.12	0.25 ± 0.28	0.18 ± 0.25	1.71 (0.18)	0	
	10 masl	0.18 ± 0.21	0.18 ± 0.23	0.11 ± 0.14			
	*F (p)*	0.64 (0.42)			Interaction *F (p-value); ω_p_*^2^* (rating)*		
	*ω_p_*^2^* (rating)*	0			1.01 (0.37); 0		
RD_HighSprinting_ (>24 km/h) (m/min)	2320 masl	0.02 ± 0.02	0.07 ± 0.09	0.06 ± 0.10	2.17 (0.12)	0	
	10 masl	0.03 ± 0.05	0.03 ± 0.07	0.02 ± 0.06			
	*F (p)*	2.62 (0.11)			Interaction *F (p-value); ω_p_*^2^* (rating)*		
	*ω_p_*^2^* (rating)*	0			1.77 (0.17); 0		
**HR_AVG_** **(bpm)**	2320 masl	137.01 ± 10.94	132.83 ± 9.61	141.01 ± 11.25	**4.04 (0.02)**	**0.03 (low)**	**f**
	10 masl	128.27 ± 12.39	127.97 ± 14.84	132.26 ± 11.77			
	*F (p)*	**13.08 (<0.01)**			Interaction *F (p value); ω_p_*^2^* (rating)*		
	*ω_p_*^2^* (rating)*	**0.06 (moderate)**			0.52 (0.60); 0		
HR_MAX_ (bpm)	2320 masl	170.69 ± 13.59	175.67 ± 10.91	178.03 ± 11.15	1.26 (0.29)	0	
	10 masl	172.36 ± 16.31	174.79 ± 14.96	174.35 ± 13.56			
	*F (p)*	0.18 (0.67)			Interaction *F (p-value); ω_p_*^2^* (rating)*		
	*ω_p_*^2^* (rating)*	0			0.40 (0.67); 0		
**50–60%HR_MAX_** **(%)**	2320 masl	8.60 ± 6.29	18.30 ± 11.96	11.14 ± 13.06	**6.74 (<0.01)**	**0.05 (low)**	**c d**
	10 masl	20.88 ± 9.08	23.06 ± 9.81	18.51 ± 9.32			
	*F (p)*	**21.89 (<0.01)**			Interaction *F (p-value); ω_p_*^2^* (rating)*		
	*ω_p_*^2^* (rating)*	**0.10 (moderate)**			1.54 (0.22); 0		
60–70%HR_MAX_ (%)	2320 masl	23.87 ± 12.19	24.46 ± 8.70	20.76 ± 9.26	0.26 (0.77)	0	
	10 masl	24.52 ± 6.94	21.81 ± 6.68	27.56 ± 12.84			
	*F (p)*	1.03 (0.31)			Interaction *F (p-value); ω_p_*^2^* (rating)*		
	*ω_p_*^2^* (rating)*	0			2.79 (0.09); 0		
**70–80%HR_MAX_** **(%)**	2320 masl	24.07 ± 5.53	20.55 ± 9.39	23.43 ± 9.59	1.96 (0.14)	0	
	10 masl	18.64 ± 7.87	17.30 ± 7.65	19.31 ± 6.12			
	*F (p)*	**10.15 (<0.01)**			Interaction *F (p-value); ω_p_*^2^* (rating)*		
	*ω_p_*^2^* (rating)*	**0.05 (low)**			0.22 (0.80); 0		
**80–90%HR_MAX_** **(%)**	2320 masl	26.67 ± 12.57	20.47 ± 9.95	24.54 ± 10.18	**5.11 (0.01)**	**0.04 (low)**	**a f**
	10 masl	17.50 ± 10.41	13.90 ± 9.43	19.90 ± 11.03			
	*F (p)*	**16.12 (<0.01)**			Interaction *F (p-value); ω_p_*^2^* (rating)*		
	*ω_p_*^2^* (rating)*	**0.08 (moderate)**			0.49 (0.61); 0		
**90–95%HR_MAX_** **(%)**	2320 masl	8.44 ± 6.82	7.54 ± 6.24	13.15 ± 7.67	**5.18 (0.01)**	**0.04 (low)**	**e f**
	10 masl	6.05 ± 5.09	5.28 ± 6.40	7.14 ± 7.60			
	*F (p)*	**10.48 (<0.01)**			Interaction *F (p-value); ω_p_*^2^* (rating)*		
	*ω_p_*^2^* (rating)*	**0.05 (low)**			1.40 (0.25); 0		
**95–100%HR_MAX_** **(%)**	2320 masl	4.89 ± 8.28	4.17 ± 5.35	5.44 ± 5.55	0.18 (0.84)	0	
	10 masl	3.15 ± 4.39	2.65 ± 5.26	1.57 ± 2.50			
	*F (p)*	**7.98 (0.01)**			Interaction *F (p-value); ω_p_*^2^* (rating)*		
	*ω_p_*^2^* (rating)*	**0.04 (low)**			0.91 (0.41); 0		
Speed_MAX_ (km/h)	2320 masl	18.76 ± 1.32	19.50 ± 1.89	19.59 ± 2.08	0.94 (0.39)	0	
	10 masl	19.41 ± 2.39	19.31 ± 3.16	20.21 ± 4.60			
	*F (p)*	0.62 (0.43)			Interaction *F (p-value); ω_p_*^2^* (rating)*		
	*ω_p_*^2^* (rating)*	0			0.48 (0.62); 0		
**Speed_AVG_** **(km/h)**	2320 masl	4.29 ± 0.39	4.18 ± 0.36	3.98 ± 0.33	**5.08 (0.01)**	**0.04 (low)**	**b d**
	10 masl	4.34 ± 0.56	4.31 ± 0.60	4.04 ± 0.53			
	*F (p)*	0.94 (.33)			Interaction *F (p-value); ω_p_*^2^* (rating)*		
	*ω_p_*^2^* (rating)*	0			0.11 (0.89); 0		
TImpacts (n/min)	2320 masl	89.24 ± 16.45	85.74 ± 14.75	90.16 ± 16.64	0.70 (0.50)	0	
	10 masl	81.42 ± 19.84	83.25 ± 21.91	86.75 ± 23.54			
	*F (p)*	2.12 (0.15)			Interaction *F (p-value); ω_p_*^2^* (rating)*		
	*ω_p_*^2^* (rating)*	0			0.24 (0.78); 0		
TImpacts_Low_ (0–3 g) (n/min)	2320 masl	77.64 ± 15.04	77.70 ± 14.20	81.39 ± 16.37	1.17 (0.31)	0	
	10 masl	70.50 ± 17.69	74.67 ± 19.21	78.03 ± 20.93			
	*F (p)*	2.51 (0.12)			Interaction *F (p-value); ω_p_*^2^* (rating)*		
	*ω_p_*^2^* (rating)*	0			0.19 (0.83); 0		
**TImpacts_Moderate_** **(3–5 g)** **(n/min)**	2320 masl	8.67 ± 1.92	6.14 ± 1.78	6.54 ± 1.60	**9.15 (<0.01)**	**0.08 (moderate)**	**a b**
	10 masl	7.96 ± 2.61	6.54 ± 2.82	6.65 ± 2.70			
	*F (p)*	0.03 (0.87)			Interaction *F (p-value); ω_p_*^2^* (rating)*		
	*ω_p_*^2^* (rating)*	0			0.70 (0.50); 0		
**TImpacts_High_** **(5–8 g)** **(n/min)**	2320 masl	2.65 ± 0.74	1.69 ± 0.97	2.02 ± 1.03	**9.07 (<0.01)**	**0.08 (moderate)**	**a b**
	10 masl	2.65 ± 1.12	1.85 ± 1.11	1.89 ± 1.08			
	*F (p)*	0.01 (0.95)			Interaction *F (p-value); ω_p_*^2^* (rating)*		
	*ω_p_*^2^* (rating)*	0			0.31 (0.73); 0		
**TImpacts_VeryHigh_** **(>8 g)** **(n/min)**	2320 masl	0.28 ± 0.11	0.21 ± 0.20	0.21 ± 0.16	**5.34 (0.01)**	**0.04 (low)**	**a b**
	10 masl	0.31 ± 0.18	0.19 ± 0.15	0.17 ± 0.14			
	*F (p)*	0.17 (0.68)			Interaction *F (p-value); ω_p_*^2^* (rating)*		
	*ω_p_*^2^* (rating)*	0			0.53 (0.59); 0		
**Total Steps** **(n/min)**	2320 masl	33.99 ± 7.76	28.09 ± 4.96	28.20 ± 5.58	**4.73 (0.01)**	**0.04 (low)**	**a b**
	10 masl	30.39 ± 8.99	27.39 ± 8.63	26.93 ± 10.18			
	*F (p)*	2.16 (0.14)			Interaction *F (p-value); ω_p_*^2^* (rating)*		
	*ω_p_*^2^* (rating)*	0			0.44 (0.64); 0		
**Total Jumps** **(n/min)**	2320 masl	0.86 ± 0.43	0.74 ± 0.42	0.82 ± 0.54	0.31 (0.73)	0	
	10 masl	0.64 ± 0.33	0.65 ± 0.38	0.56 ± 0.28			
	*F (p)*	**8.39 (<0.01)**			Interaction *F (p value); ω_p_*^2^* (rating)*		
	*ω_p_*^2^* (rating)*	**0.04 (low)**			0.83 (0.44); 0		
**PL_RT_** **(a.u./min)**	2320 masl	0.70 ± 0.12	0.60 ± 0.08	0.61 ± 0.07	**5.63 (<0.01)**	**0.05 (low)**	**a b**
	10 masl	0.65 ± 0.15	0.58 ± 0.15	0.59 ± 0.15			
	*F (p)*	1.87 (0.17)			Interaction *F (p-value); ω_p_*^2^* (rating)*		
	*ω_p_*^2^* (rating)*	0			0.41 (0.67); 0		
**TAcc** **(n/min)**	2320 masl	32.24 ± 3.55	30.97 ± 2.93	32.37 ± 2.37	2.39 (0.09)	0	
	10 masl	23.60 ± 4.80	23.26 ± 5.40	25.20 ± 5.55			
	*F (p)*	**120.23 (<0.01)**			Interaction *F (p-value); ω_p_*^2^* (rating)*		
	*ω_p_*^2^* (rating)*	**0.39 (high)**			0.29 (.75); 0		
**TDec** **(n/min)**	2320 masl	32.23 ± 3.58	30.97 ± 2.91	32.42 ± 2.35	2.46 (0.08)	0	
	10 masl	33.56 ± 7.78	23.27 ± 5.40	25.23 ± 5.57			
	*F (p)*	**120.77 (<0.01)**			Interaction *F (p-value); ω_p_*^2^* (rating)*		
	*ω_p_*^2^* (rating)*	**0.39 (high)**			0.29 (0.74); 0		
**Acc_MAX_** **(m/s^2^)**	2320 masl	5.15 ± 0.41	4.96 ± 0.52	5.09 ± 0.46	0.71 (0.49)	0	
	10 masl	4.29 ± 0.98	4.35 ± 0.78	4.01 ± 0.47			
	*F (p)*	**60.23 (<0.01)**			Interaction *F (p value); ω_p_*^2^* (rating)*		
	*ω_p_*^2^* (rating)*	**0.24 (high)**			2.08 (0.13); 0		
**Dec_MAX_** **(m/s^2^)**	2320 masl	5.03 ± 0.33	4.84 ± 0.47	4.87 ± 0.40	1.80 (0.17)	0	
	10 masl	4.27 ± 0.99	4.26 ± 0.77	3.91 ± 0.44			
	*F (p)*	**52.36 (<0.01)**			Interaction *F (p value); ω_p_*^2^* (rating)*		
	*ω_p_*^2^* (rating)*	**0.21 (high)**			1.37 (.26); 0		
**RD in** **Acceleration** **(m/min)**	2320 masl	6.40 ± 1.18	5.15 ± 1.42	5.18 ± 1.41	**5.74 (<0.01)**	**0.04 (low)**	**a b**
	10 masl	4.01 ± 1.96	3.62 ± 1.78	2.78 ± 1.85			
	*F (p)*	**61.52 (<0.01)**			Interaction *F (p value); ω_p_*^2^* (rating)*		
	*ω_p_*^2^* (rating)*	**0.23 (high)**			1.50 (0.23); 0		
**RD in** **Deceleration** **(m/min)**	2320 masl	5.49 ± 1.14	4.75 ± 1.45	4.68 ± 1.44	**5.11 (<0.01)**	**0.03 (low)**	**a b**
	10 masl	3.49 ± 1.72	3.09 ± 1.53	2.19 ± 1.52			
	*F (p)*	**70.94 (<0.01)**			Interaction *F (p-value); ω_p_*^2^* (rating)*		
	*ω_p_*^2^* (rating)*	**0.26 (high)**			1.27 (0.28); 0		

**Note:** masl: meters above sea level; HR: heart rate; PL: player load; RD: relative distance; F: F-value of MANOVA; *p*: *p*-value; *ω_p_*^2^**: partial omega squared. **Bold font represents statistical differences**. Post hoc: (a) guards > forwards; (b) guards > centers; (c) forwards > guards; (d) forwards > centers; (e) centers > guards; (f) centers > forwards.

## Data Availability

The data presented in this study are available upon request from the corresponding author. The data are not publicly available due to the Organic Law 3/2018, of 5 December, on the Protection of Personal Data and Guarantee of Digital Rights of the Government of Spain, which requires that this information must be in custody.

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
