# Peer review of "Examining the Effects of Altitude on Workload Demands in Professional Basketball Players during the Preseason Phase"

_sensors, 2024, doi:10.3390/s24103245_

Round 1

Reviewer 1 Report

Comments and Suggestions for Authors

Dear Author

Congratulations  on the work, however I would like to lightning some points as a suggestion:

In abstract: I suggest following the magazine's guideline (style of structured abstract).

In references: I suggest that all references follow the same format. Please follow the journal's guidelines.

Author Response

Reply letter of Reviewer 1

Reviewer 2 Report

Comments and Suggestions for Authors

Your paper needs  several clarifications and in addition a clear statement in the limitations in respect to a long merit effect of high altitude training or you provide additional data to show that training at high altitude at least  lasts for some period of time.

See comments highlighted in your submission.

From the point of view of the topic your paper does not adress "Artificial Intelligence (AI) and Sensors in Sports Safety and NextGen Rehabilitation".

Comments on the Quality of English Language

English is fine

Author Response

Reply letter of Reviewer 2

Reviewer 3 Report

Comments and Suggestions for Authors

Reviewer Comments:

The study offers valuable insights into the effects of altitude training on the workloads of professional basketball players during the preseason, contributing significantly to the sports science field.  However, to further enhance the study's clarity and comprehensiveness, certain revisions and improvements are recommended.

1.      The introduction could benefit from a clear definition of "internal load" and "external load" to avoid ambiguity and ensure a comprehensive understanding for readers.

2.      The discussion section should focus exclusively on the study's findings. Currently, it includes references to performance improvements after returning to sea level, which is not directly supported by the study's results, leading to potential confusion.

3.      The study's limitation section could benefit from suggesting future research to analyze differences in physical strength between short-term block altitude training and sea-level training, expanding the study's scope and applicability.

4.      The study provides a valuable contribution to sports science, particularly in understanding altitude training's effects on basketball workloads. Incorporating these suggested changes would strengthen the study's clarity and focus, enhancing its impact and relevance.

Author Response

reply letter of reviewer 3

Round 2

Reviewer 2 Report

Comments and Suggestions for Authors

Nicely improved and all questions with one exemption answered, namely why the multiplication by factor 0,01 in formula for PLRT ?

Comments on the Quality of English Language

no comment

Author Response

Round 2 of Reviewer 2
